# Learning Category Trees for ID-Based Recommendation: Exploring the Power of Differentiable Vector Quantization

Submission Id: 1037*

## ABSTRACT

Category information plays a crucial role in enhancing the quality and personalization of recommender systems. Nevertheless, the availability of item category information is not consistently present, particularly in the context of ID-based recommendations. In this work, we propose a novel approach to automatically learn and generate entity (i.e., user or item) category trees for ID-based recommendation. Specifically, we devise a differentiable vector quantization framework for automatic category tree generation, namely CAGE, which enables the simultaneous learning and refinement of categorical code representations and entity embeddings in an end-to-end manner, starting from the randomly initialized states. With its high adaptability, CAGE can be easily integrated into both sequential and non-sequential recommender systems. We validate the effectiveness of CAGE on various recommendation tasks including list completion, collaborative filtering, and click-through rate prediction, across different recommendation models. We release the code and data[1] for others to reproduce the reported results.

## CCS CONCEPTS

• **Information systems** → **Recommender systems**; **Clustering and classification**; *Data mining*.

## KEYWORDS

recommender system, differentiable vector quantization

### ACM Reference Format:

Anonymous Author(s). 2018. Learning Category Trees for ID-Based Recommendation: Exploring the Power of Differentiable Vector Quantization. In *Proceedings of Make sure to enter the correct conference title from your rights confirmation emai (Conference acronym 'XX).* ACM, New York, NY, USA, 10 pages. https://doi.org/XXXXXXX.XXXXXXX

## 1 INTRODUCTION

Recommender systems [15, 42, 48] aim to ease the burden of decision-making by automatically suggesting personalized item recommendations tailored to a user's preferences and historical behavior. They cater to diverse objectives such as list completion, collaborative filtering, and click-through rate prediction. The varied objectives underscore the importance of devising methodologies that can

[1]https://anonymous.4open.science/r/TheWebConf24-Cage/

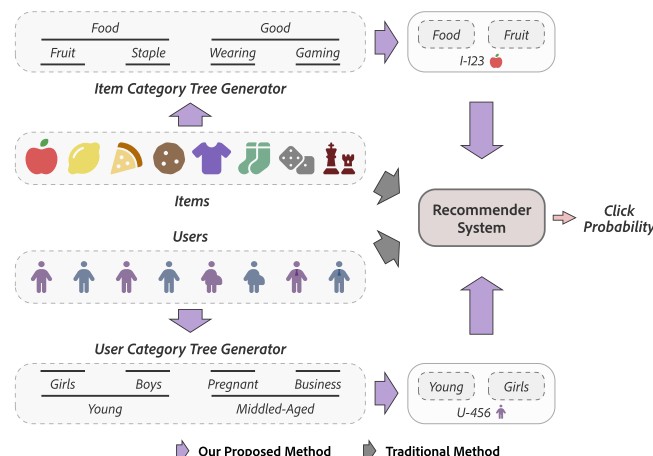

**Figure 1: Illustration of our approach in learning category trees for ID-based recommendation. In contrast to traditional methods that solely offer item or user IDs to the recommender system, our approach involves implicit learning of user/item category trees. The category information, encoded as vectors, is subsequently integrated with the user/item ID and provided as input to the recommender system.**

adapt to different recommendation scenarios and deliver improved recommendations.

When crafting recommendation models and algorithms, the integration of categorical information is of paramount importance. Categorical attributes, such as product types [6] and user locations [29, 34], find widespread use due to their ability to capture crucial attributes and establish meaningful connections for users and items. Furthermore, these category features serve to mitigate the cold-start problem, providing an additional layer of information for less active (sparsely interacting) entities (i.e., users or items) [2, 11]. This supplementary information is progressively refined by interactions from active users or items during training, thereby aiding less active entities in obtaining more robust representations.

However, category features are not always available, since many recommendation datasets only have ID information. To address the absence of category attributes in ID-based recommendation contexts, we propose a novel automatic **ca**tegory tree **ge**neration framework, namely CAGE. As illustrated in Figure 1, CAGE serves as the precursor to the recommender system, dynamically constructing an item/user category tree, which incorporates hierarchical categorical knowledge (e.g., "Good" and "Gaming") relevant to the current entity (e.g., "I-123"). The categorical information, encoded as vectors, is provided to the recommender system as auxiliary information alongside the user/item ID. The implementation of CAGE is based on differentiable and cascaded vector quantization (VQ).

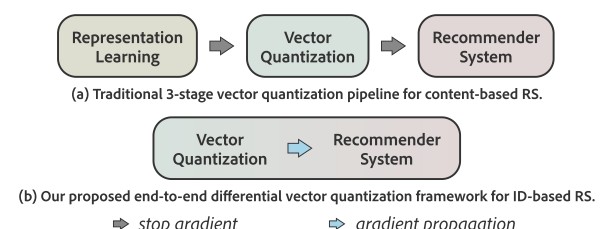

(a) Traditional 3-stage vector quantization pipeline for content-based RS.

(b) Our proposed end-to-end differential vector quantization framework for ID-based RS.

⇨ *stop gradient*      ⇨ *gradient propagation*

**Figure 2: Comparison between (a) the traditional three-stage vector quantization pipeline for content-based recommendation and (b) our proposed end-to-end differential vector quantization framework for ID-based recommendation.**

Previous vector quantization methods for recommendation [37, 52] often rely on *meaningful and fixed* entity (e.g., user or item) embeddings, derived from side information like content-aware item embeddings using pretrained models. They commonly adopt a three-stage design, as displayed in Figure 2, where representation learning, vector quantization, and recommendation training are carried out separately. However, the lack of side information in ID-based recommendation hinders the generation of meaningful entity embeddings during the initial training phase, making this approach impractical. We tackle this challenge by employing differentiable VQ [46]. It enables dynamic adjustments of both entity embeddings and categorical code vectors from the quantization codebooks through recommendation tasks and quantization constraints, starting from their initial random states and resulting in a robust and stable form.

Moreover, it is crucial to select the appropriate level of detail for categories. Employing finely detailed categories could potentially result in data sparsity issues within the recommendation system, while adopting broader, coarse-grained categories may obscure significant differentiations among entities. In light of this challenge, we propose cascaded VQ to construct a category tree with varying levels of granularity. Unlike the single-layer category system commonly used in datasets, our cascaded approach creates a hierarchical taxonomy of categories, offering a more comprehensive representation of entities.

To summarize, we introduce CAGE, an automatic category tree generation framework for ID-based recommendation, which offers several notable advantages and capabilities as outlined below.

- **End-to-end framework.** Differing from the common multi-stage application of VQ in recommender systems [37, 52], we are the first to explore differentiable VQ as an end-to-end solution for category generation in a scenario without side information, i.e., ID-based recommendation. Such end-to-end training allows for refining and optimizing the categorization for both items and users to align with specific recommendation objectives.
- **Easy adoption and high adaptability.** CAGE is a pluggable module that can be conveniently integrated into both sequential and non-sequential recommendation models for accommodating different recommendation scenarios, including list completion, collaborative filtering, and click-through rate prediction.

- **Effectiveness.** We conduct a comprehensive evaluation of CAGE on multiple recommendation tasks, including list completion, collaborative filtering, and click-through rate prediction. The evaluation involves seven datasets and a comparison with 14 baseline methods. The results demonstrate the effectiveness of CAGE, showcasing significant improvements across most scenarios. For example, CAGE demonstrates a relative improvement of up to 21.41% over state-of-the-art baselines in list completion tasks and up to 37.08% in collaborative filtering tasks.

## 2 RELATED WORK

### 2.1 Vector Quantization

Vector quantization (VQ) techniques [12] map a large set of input vectors into a small set of vectors (i.e., a codebook), which have been widely studied in computer vision [1, 38, 51] and speech coding [5, 22] domains. To date, only a few studies explore the potential of vector quantization in recommendation systems. One line of research aims to improve recommendation efficiency [24, 28, 45]. The other line of research focuses on improving recommendation quality [35, 37], as shown by the growing interest from researchers in recent years. The studies on enhancing recommendation quality can be categorized into two paradigms: the commonly used multi-stage approach [19, 37, 52] and an end-to-end training [35] strategy.

To our knowledge, AQCL [35] is the only work using end-to-end training for quality improvement in ID-based recommendation, which leverages VQ to assist contrastive learning in the CTR prediction scenario. Differ from AQCL, our proposed CAGE, is the first to introduce VQ for learning categorical knowledge in ID-based recommendation.

### 2.2 Recommender Systems

Recommender systems have been extensively studied in various application scenarios including (1) list completion, which aims to continue the user-curated list by sequence generation, (2) collaborative filtering (CF) that makes recommendation based on user-item interactions, and (3) click-through rate (CTR) prediction, which is a crucial task in the ranking phase of the recommendation pipeline.

**List completion.** Pioneer works based on Markov chain [8, 32, 33] or neural networks [7, 10, 44, 47] are mostly proposed for automatic playlist continuation. In recent years, sequential recommenders [17, 18, 42, 43] have been proposed to generation items autoregressive for list completion task, while FANS [30] uses non-autoregressive generation to improve both quality and efficiency.

**Collaborative filtering.** Collaborative filtering (CF) is widely adopted in the matching phase of the recommendation pipeline. Traditional CF methods [4, 25, 27, 40] employ neighborhood-based approaches and use similarity metrics to identify users or items with similar preferences, which face scalability and sparsity issues in large-scale systems. To overcome these limitations, matrix factorization [26] techniques have been widely adopted to capture underlying preferences and characteristics for personalized recommendation. More recently, deep learning-based methods [15, 16, 39, 53] have emerged to learn complex user-item interactions and capture nonlinear relationships.

**Click-through rate prediction.** In recent years, deep learning-based CTR prediction models [9, 13, 20, 31, 48] have gained popularity. These models have demonstrated improved performance by leveraging the expressive power of neural networks to capture intricate patterns in user-item interactions.

## 3 PROPOSED FRAMEWORK: CAGE

Figure 3 illustrates our automatic category tree generation (CAGE) framework, which is designed to enhance *id-based* representations of both items and users. It involves a series of cascaded vector quantizers for extracting category-aware information at multiple levels of granularity. More precisely, the vector quantizers are interconnected in a successive manner, with the output of one quantizer be the input to the next. The quantized multi-level code vectors are then fused and fed to the recommender system to facilitate downstream recommendation tasks. CAGE and the recommender system are trained together in an end-to-end manner.

### 3.1 Tree Construction

Before training, both the embedding vectors and code vectors in codebooks are randomly initialized. There are no connections between each entity and the nodes in the bottom-level codebook (the leftmost layer in Figure 3), as well as between adjacent codebooks.

*3.1.1 Searching with Vector Quantization.* Specifically, we establish links between adjacent layers using the vector quantization technique. Vector quantization [50] targets at grouping similar vectors into clusters by representing them with a small set of prototype vectors. We use a vector quantizer to locate the code vector within a codebook that closely matches the input embedding. The code vector is anticipated to capture and represent the categorical information associated with the input embedding. The vector quantizer includes a $k$-entry codebook $\mathbf{C} \in \mathbb{R}^{k \times d}$, where $k$ is the number of the code vectors and $d$ is the dimension of each code vector.

Given an input embedding $\mathbf{e} \in \mathbb{R}^d$, nearest neighbour search is performed to find the most similar code to $\mathbf{z}$ within $\mathbf{C}$:

$$j = \arg \min_{i \in \{1,2,\ldots,k\}} \| \mathbf{e} - \mathbf{c}_i \|_2^2, \tag{1}$$

where $\mathbf{c}_i (1 \le i \le k)$ is any code vector in the codebook $\mathbf{C}$, and $j$ is the index of the matched code vector $\mathbf{c}_j$. Note that the current matching pair $(\mathbf{e}, \mathbf{c}_j)$ is a **temporary result** that evolves as the training process continually adjusts entity embeddings and code vectors.

*3.1.2 Cascaded Linking Flow.* CAGE employs a series of cascaded vector quantizers to capture categorical information at multiple levels of granularity. Figure 3 shows an example with three quantizers.

Let $H$ be the number of quantizers (or levels of granularity). Each quantizer $Q^{(i)}$ has a $v^i$-entry codebook $\mathbf{C}^{(i)}$, where $i = 1, 2, \ldots, H$. The quantizers are interconnected in a cascaded fashion, generating *fine-to-coarse* code vectors, i.e., $v^i > v^j$ for $i < j$. Each quantizer $Q^{(i)}$ takes the output of the previous quantizer (i.e., $Q^{(i-1)}$) as input, creating a quantization flow defined as follows.

$$\mathbf{c}^{(i)} = Q^{(i)} \left( \mathbf{c}^{(i-1)} \right), \tag{2}$$

$$\mathbf{c}^{(0)} = \mathbf{e}, \tag{3}$$

where $\mathbf{c}^{(i)}$ is the output of quantizer $Q^{(i)}$.

### 3.2 Code Fusion Layer

After obtaining multi-level codes (or categories) $\mathbf{c}_q^{(i)} (i = 1, 2, \cdots, H)$, we employ an average pooling operation to combine them into a single vector:

$$\bar{\mathbf{c}} = \frac{1}{H} \sum_i^H \mathbf{c}^{(i)}. \tag{4}$$

In addition, we use a weighted residual connection to add the original vector $\mathbf{e}$ to obtain the final category-aware representation $\mathbf{z}$:

$$\mathbf{z} = \mathbf{e} + \alpha \bar{\mathbf{c}}, \tag{5}$$

where $\alpha$ is a hyperparameter that balances the two terms. We use "$f$" to denote the aforementioned operations, i.e., $\mathbf{z} = f(\mathbf{e})$.

### 3.3 Tree Back Propagation

Since the nearest neighbour search algorithm is not differentiable, we utilize the straight-through estimator (STE) [3] to approximate the gradient of each quantizer. Specifically, the gradient of the quantizer is approximated by the gradient of the identity function, which is defined as:

$$\frac{\partial \mathbf{c}^{(i)}}{\partial \mathbf{c}^{(i-1)}} \approx \frac{\partial \mathbf{c}^{(i-1)}}{\partial \mathbf{c}^{(i-1)}} = \mathbf{I}, \tag{6}$$

where $\mathbf{I}$ is the identity matrix. Therefore, the quantization loss (encouraging the quantizer to select the closest vector in the codebook) can be defined as:

$$L_{\text{quant}} = \sum_i^H \left( \| sg[\mathbf{c}^{(i-1)}] - \mathbf{c}^{(i)} \|_2^2 \right), \tag{7}$$

where $sg$ is the stop gradient operation. Furthermore, we introduce a commitment loss that encourages the input embedding $\mathbf{c}^{(i-1)}$ to approach the currently matched code vector $\mathbf{c}^{(i)}$, which reduces the frequency of link changes, resulting in a smoother training process:

$$L_{\text{commit}} = \sum_i^H \left( \| \mathbf{c}^{(i-1)} - sg[\mathbf{c}^{(i)}] \|_2^2 \right). \tag{8}$$

Finally, the overall tree generation loss can be defined by:

$$L_{\text{cage}} = L_{\text{quant}} + \beta L_{\text{commit}}, \tag{9}$$

where $\beta$ is a hyper-parameter that controls the trade-off between the two losses.

### 3.4 End-to-end Training

As mentioned in Section 3.1, the cascaded code vectors and entity embeddings are both initialized randomly prior to training. Initially, entity embeddings lack meaningful information, leading to insignificant quantization outcomes. As training progresses, the CAGE module and the recommender model are jointly optimized through an external recommendation task (i.e., recommendation loss), gradually imbuing entity embeddings with semantic context. Furthermore, internal tree generation loss, $L_{\text{cage}}$, is introduced to enhance the clustering effectiveness of the codebook. The enriched category information (code representation) subsequently contributes to improved recommendation performance for entity

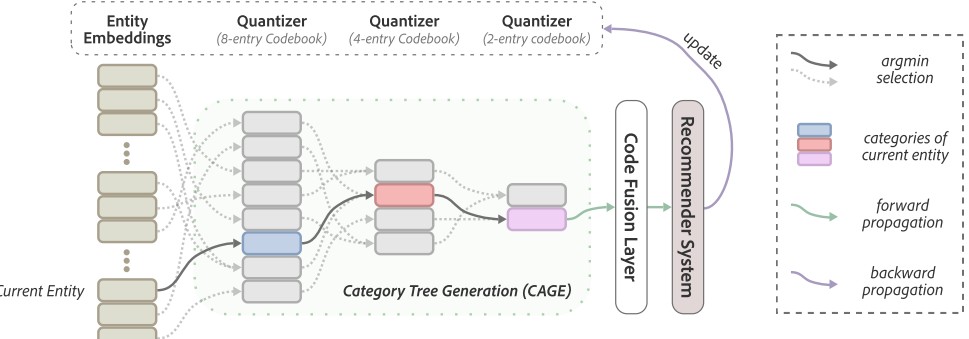

**Figure 3: Overview of our proposed category tree generation framework (CAGE).**

embeddings in subsequent training batches. This cyclic iteration results in a double-helix refinement process, where the codebook and entity embeddings continuously enhance their representation learning throughout the training process.

## 4 APPLICATIONS

CAGE can be effortlessly and seamlessly integrated into a variety of recommendation models to enhance recommendation performance, making it highly adaptable and suitable for a wide range of recommendation scenarios. In the following, we demonstrate how CAGE can be applied to non-sequential (e.g., collaborative filtering and CTR prediction) and sequential recommenders (e.g., list completion).

### 4.1 CAGE for Non-sequential Recommenders: Incorporating both Item and User CAGE

*4.1.1 Scenario 1: Collaborative Filtering.* Given a set of users $\mathcal{U}$ and a set of items $\mathcal{V}$, the collaborative filtering agent [15, 16, 39, 53] aims to estimate the user-item interaction matrix $\mathbf{R}$, where each weight $r_{ui}$ represents the preference or rating of user $u$ for item $i$. The matrix $\mathbf{R}$ is typically sparse, as not all users rate or interact with all items. Therefore, the task is to fill in these missing weights by existing ratings.

*4.1.2 Scenario 2: Click-Through Rate Prediction.* In contrast to collaborative filtering, which is typically employed in the matching phase of the recommender pipeline, CTR prediction is a ranking-based task that aims to predict the probability of a user clicking on a particular item. The input to the CTR model is a user-item pair, and the output is a probability score indicating the likelihood of the user clicking on the item. Existing deep CTR prediction models [13, 20, 36, 48] are typically designed to learn feature interactions from the raw input features such as user ID, item ID, and other statistical features if exists.

*4.1.3 Integration.* In the non-sequential scenario, both user and item representations are obtained as embedding matrices and play integral roles in the training process. Therefore, we can incorporate two CAGE modules (i.e., Item CAGE $f^{(i)}$ and User CAGE $f^{(u)}$) on

dual sides to extract hierarchical category knowledge, denoted as:

$$\mathbf{z}^{(u)} = f^{(u)}\left(\mathbf{e}^{(u)}\right), \mathbf{z}^{(i)} = f^{(i)}\left(\mathbf{e}^{(i)}\right). \tag{10}$$

Finally, the loss function can be calculated by:

$$L_{\text{rec}} = \Phi\left(\mathbf{z}^{(u)}, \mathbf{z}^{(u)}, l\right), \tag{11}$$

$$L = L_{\text{rec}} + \omega_q\left(L_{\text{cage}}^{(u)} + L_{\text{cage}}^{(i)}\right), \tag{12}$$

where $\Phi$ is the non-sequential recommender, $l$ is the label, and $\omega_q$ is a hyperparameter balancing the internal tree generation loss and external recommendation loss.

### 4.2 CAGE for Sequential Recommenders: Incorporating Item CAGE with an Additional Category Tree Classification Loss

*4.2.1 Scenario: List Completion.* Given a set of item vocabulary $\mathcal{V}$ ($v^0 = |\mathcal{V}|$) and a user curated list $\mathbf{x} = [x_1, x_2, \cdots, x_{|\mathbf{x}|}]$ ($x_i \in \mathcal{V}$), the list completion agent [17, 23, 30, 43] is required to predict an item sequence $\mathbf{y} = [y_1, y_2, \cdots, y_{|\mathbf{y}|}]$ ($y_i \in \mathcal{V}$) that is a subsequent of $\mathbf{x}$, which can be formulated as maximizing the probability

$$p\left(\mathbf{y}' = \mathbf{y}|\mathbf{x}\right), \tag{13}$$

where $\mathbf{y}'$ represents any possible list of length $|\mathbf{y}|$.

Different from non-sequential recommenders which generate a scalar score, the output of the list completer is a prediction item vector $\bar{\mathbf{z}}$. More precisely, the item completer undergoes training using the item prediction task[2]. Therefore, a classification module is designed to infer the probability distribution over the item vocabulary by the softmax function for each prediction item vector:

$$g^0 : \mathbb{R}^d \to \mathbb{R}^{v^0}. \tag{14}$$

*4.2.2 Integration.* In the sequential scenario, since the user representation is derived by fusing the historical item list, we only need to insert a single CAGE module for item categorization. Additionally, the category tree generated by our CAGE naturally serves as a valuable aid for the item prediction task.

Assuming that the ground truth label for the item vector to be predicted, $\bar{\mathbf{z}}$, is the $y^{(0)}$-th item, and its current embedding is $\mathbf{z}$. We can start by using Item CAGE to obtain pseudo-labels (i.e., ***code***

---

[2]This task can take the form of predicting the next item for autoregressive methods [17, 43] or a masked item prediction task [30] for non-autoregressive methods.

| Datasets | List Completion | | | CTR | | CF | |
|---|---|---|---|---|---|---|---|
| | Zhihu | Spotify | Goodreads | MIND | MovieLens | Toys | Kindle |
| #Users | 18,704 | 72,152 | 15,426 | 94,057 | 943 | 19,413 | 68,224 |
| #Items | 36,005 | 104,695 | 47,877 | 65,238 | 1,682 | 11,925 | 61,935 |
| #Interactions | 927,781 | 6,809,820 | 1,589,480 | 1,756,555 | 52,480 | 623,023 | 2,664,795 |
| Items per list | 49.59 | 94.38 | 103.04 | - | - | - | - |
| List Range | $10 \sim 200$ | $20 \sim 300$ | $20 \sim 300$ | - | - | - | - |
| Samples | - | - | - | 9,993,270 | 69,881 | 1,246,064 | 5,329,590 |
| Density | 0.138% | 0.089% | 0.215% | 0.163% | 4.406% | 0.538% | 0.136% |

Table 1: Dataset statistics. The density is defined as the ratio of the number of interactions to the number of all possible interactions.

*indices* in the multi-level codebooks, denoted as $y^{(i)}$, $i = 1, 2, \ldots, H$ for the current embedding $\mathbf{z}$. Subsequently, we design an auxiliary *tree classification task* that encourages the current prediction vector $\bar{\mathbf{z}}$ to predict the category it corresponds to in each layer of the category tree which multiple node classification module:

$$g^i : \mathbb{R}^d \to \mathbb{R}^{v^i}, i = 1, 2, \ldots, H. \tag{15}$$

Such auxiliary task further strengthens the connection between items and categories, leading to more precise predictions. Then, we proceed with the multi-level classification training and the loss function can be defined as:

$$L_{\text{item}} = g^0 (\mathbf{z})_{y^{(0)}}, \tag{16}$$

$$L_{\text{tree}} = \frac{1}{H} \sum_i^H g^i (\mathbf{z})_{y^{(i)}}. \tag{17}$$

Finally, the overall recommendation loss function is:

$$L_{\text{rec}} = L_{\text{item}} + \omega_c L_{\text{tree}}, \tag{18}$$

$$L = L_{\text{rec}} + \omega_q L_{\text{cage}}, \tag{19}$$

where $\omega_c$ is a hyperparameter that controls the importance of the tree classification loss, and please refer to Equation 12 for $\omega_q$.

# 5 EXPERIMENT

## 5.1 Experimental Setup

*5.1.1 Datasets.* We conducted offline experiments on three recommendation tasks, namely list completion, collaborative filtering (CF), and click-through rate (CTR) prediction. For the list completion task, we use three real-world datasets: Zhihu, Spotify, and Goodreads, which were crawled and compiled by [17]. For the collaborative filtering task, we utilize two public datasets: Amazon Toys and Amazon Kindle Store, namely Toys and Kindle, respectively. Regarding the CTR prediction task, we employ two public datasets: MIND [49] (small version) and MovieLens [14] (100K version). The dataset statistics can be found in Table 1.

*5.1.2 Preprocessing.* For the list completion task, we adopt the data preprocessing steps proposed by [30]. We iteratively perform the following two operations until the data no longer changes: 1) remove items with a frequency less than 10 from all lists; 2) truncate or filter the item list according to the maximum and minimum lengths specific to each dataset. Furthermore, we uniformly divide a qualifying list into two segments, namely the input and target

lists. The lists are then partitioned into training, validation, and testing sets using an 8:1:1 ratio.

For the CF and CTR prediction datasets, we only use the user-item interaction data without any additional information. To be specific, for the MIND dataset, user historical behaviors are transformed into a list of user-item pairs, which are subsequently included in the training set. More details about the dataset preprocessing will be provided in the public code repository upon accepted.

*5.1.3 Baselines and Variants of Our Method.* **List completion.** We take the state-of-the-art sequential recommendation methods and the item list completion models as baselines, including Caser [43], GRU4Rec [18], SASRec [23], BERT4Rec [42], CAR [17] and FANS [30]. We integrate CAGE into BERT4Rec and FANS to obtain $\text{CAGE}_{\text{BERT4Rec}}$ and $\text{CAGE}_{\text{FANS}}$ models, respectively.

It is worth noting that FANS [30] pre-extracts categorical item features based on the curated item lists among training, validation, and testing sets. These categorical knowledge is also added into baseline models for a fair comparison in the FANS paper. Since we learn the cascaded categorical features in an end-to-end manner, we do not use the pre-extracted categorical entity features in our experiments for both our method variants and baselines.

**Collaborative filtering.** We compare our method with representative CF models as baselines, including BPRMF [39], NeuMF [16], CFKG [53] and LGCN [15] We integrate our proposed CAGE module into these baselines and denote them as $\text{CAGE}_{\text{BPRMF}}$, $\text{CAGE}_{\text{NeuMF}}$, $\text{CAGE}_{\text{CFKG}}$, and $\text{CAGE}_{\text{LGCN}}$, respectively.

**Click-through rate prediction.** We compare our method with the widely used and state-of-the-art deep CTR models, including DeepFM [13], DCN [48], FiBiNET [20], and FinalMLP [31]. We integrate our proposed CAGE module into these baselines and denote the integrated models as $\text{CAGE}_{\text{DeepFM}}$, $\text{CAGE}_{\text{DCN}}$, $\text{CAGE}_{\text{FinalMLP}}$, and $\text{CAGE}_{\text{FiBiNET}}$, respectively.

*5.1.4 Evaluation Protocols.* We follow the common practice [41] to evaluate the effectiveness of recommendation models with the widely used metrics, i.e., Normalized Discounted Cumulative Gain [21] (NDCG@k) and Hit Ratio (HR@k). In this work, we set $k = \{5, 10\}$.

*5.1.5 Implementation Details.* During training, we adopt the Adam optimizer as the gradient descent algorithm. For all models, the embedding dimension is set to 64. **For the list completion task**, we set the batch size to 256 and the learning rate to 0.01 following [30]. We use 3 Transformer layers for all Transformer-based models and 3 hidden layers for the GRU4Rec model. For the Caser model, we follow the original implementation and settings, and set the max sequence length to 5. We set the number of attention heads to 8 for all Transformer-based methods on the three datasets of list completion. **For the collaborative filtering task**, we set the batch size to 1024 and the learning rate to 0.001. For the LGCN model, we set the number of GCN layers to 3. **For the CTR prediction task**, we set the batch size to 5000, the learning rate to 0.001, the number of DNN layers to 3, the size of each hidden layer to 1000, and the dropout rate to 0.1 for all models. For the DCN model, we set the number of cross layers to 3. For the FiBiNET model, we set the number of feature interaction blocks to 3.

We carefully tune the hyper-parameters of all models on the validation set and report the best results achieved on the test set.

| Models | | Caser (2018) | GRU4Rec (2016) | SASRec (2018) | CAR (2020b) | BERT4Rec (2019) | CAGE$_{BERT4Rec}$ (ours) | FANS* (2023) | FANS$_{TSC}$* (2023) | CAGE$_{FANS}$ (ours) | Imp. |
|---|---|---|---|---|---|---|---|---|---|---|---|
| Zhihu | N@5 | 0.0065 | 0.0058 | 0.0046 | 0.0050 | 0.0136 | **0.0220** | 0.0256 | 0.0232 | **0.0301** | 17.58% |
| | N@10 | 0.0105 | 0.0085 | 0.0074 | 0.0087 | 0.0198 | **0.0305** | 0.0389 | 0.0337 | **0.0428** | 10.03% |
| | HR@5 | 0.0926 | 0.0819 | 0.0728 | 0.0770 | 0.1664 | **0.2333** | 0.2857 | 0.2670 | **0.3034** | 6.20% |
| | HR@10 | 0.1812 | 0.1597 | 0.1423 | 0.1664 | 0.2933 | **0.3987** | 0.4819 | 0.4604 | **0.4859** | 0.83% |
| Spotify | N@5 | 0.0187 | 0.0041 | 0.0037 | 0.0040 | 0.0136 | **0.0202** | 0.0313 | 0.0315 | **0.0352** | 11.75% |
| | N@10 | 0.0262 | 0.0057 | 0.0054 | 0.0057 | 0.0229 | **0.0298** | 0.0461 | 0.0438 | **0.0519** | 12.58% |
| | HR@5 | 0.2786 | 0.0805 | 0.0825 | 0.0793 | 0.2350 | **0.3242** | 0.4071 | 0.3992 | **0.4385** | 7.71% |
| | HR@10 | 0.3983 | 0.1236 | 0.1257 | 0.1227 | 0.3212 | **0.4559** | 0.5927 | 0.5552 | **0.6282** | 5.99% |
| Goodreads | N@5 | 0.0039 | 0.0053 | 0.0049 | 0.0040 | 0.0108 | **0.0130** | 0.0334 | 0.0293 | **0.0399** | 19.46% |
| | N@10 | 0.0053 | 0.0068 | 0.0064 | 0.0058 | 0.0160 | **0.0180** | 0.0467 | 0.0418 | **0.0567** | 21.41% |
| | HR@5 | 0.0694 | 0.0856 | 0.0830 | 0.0726 | 0.1634 | **0.1829** | 0.3819 | 0.3268 | **0.4275** | 11.94% |
| | HR@10 | 0.1109 | 0.1252 | 0.1187 | 0.1109 | **0.2678** | **0.2678** | 0.5149 | 0.4514 | **0.5473** | 6.29% |

**Table 2: Effectiveness of CAGE in list completion. We bold the best results. Asterisk symbol * indicates that the method uses pre-extracted categorical features which are learnt from the overall dataset including the test set.**

| Models | | BPRMF (2012) | CAGE$_{BPRMF}$ (ours) | Imp. | NeuMF (2017) | CAGE$_{NeuMF}$ (ours) | Imp. | CFKG (2018) | CAGE$_{CFKG}$ (ours) | Imp. | LGCN (2020a) | CAGE$_{LGCN}$ (ours) | Imp. |
|---|---|---|---|---|---|---|---|---|---|---|---|---|---|
| Toys | N@5 | 0.2284 | **0.2352** | 2.98% | 0.1728 | **0.1798** | 4.05% | 0.1571 | **0.1823** | 16.04% | 0.2301 | **0.2345** | 1.91% |
| | N@10 | 0.2595 | **0.2702** | 4.12% | 0.2038 | **0.2119** | 3.97% | 0.1894 | **0.2173** | 14.73% | 0.2658 | **0.2698** | 1.50% |
| | HR@5 | 0.3131 | **0.3292** | 5.14% | 0.2431 | **0.2545** | 4.69% | 0.2267 | **0.2585** | 14.03% | 0.3173 | **0.3262** | 2.80% |
| | HR@10 | 0.4095 | **0.4377** | 6.89% | 0.3392 | **0.3543** | 4.45% | 0.3272 | **0.3671** | 12.19% | 0.4282 | **0.4360** | 1.82% |
| Kindles | N@5 | 0.4376 | **0.4851** | 10.85% | 0.4341 | **0.4431** | 2.07% | 0.2837 | **0.3889** | 37.08% | 0.5105 | **0.5123** | 0.35% |
| | N@10 | 0.4826 | **0.5207** | 7.89% | 0.4708 | **0.4789** | 1.72% | 0.3235 | **0.4298** | 32.86% | 0.5438 | **0.5476** | 0.70% |
| | HR@5 | 0.5903 | **0.6205** | 5.12% | 0.5646 | **0.5711** | 1.15% | 0.3891 | **0.5263** | 35.26% | 0.6466 | **0.6573** | 1.65% |
| | HR@10 | 0.7287 | **0.7299** | 0.16% | 0.6778 | **0.6816** | 0.56% | 0.5125 | **0.6529** | 27.40% | 0.7492 | **0.7660** | 2.24% |

**Table 3: Effectiveness of CAGE in collaborative filtering. We bold the best results.**

The results are averaged over 5 runs. Due to space constraints, we will furnish the details in future publications. All the methods were trained using NVIDIA GeForce RTX 3090 with 24GB memory.

## 5.2 Main Results

**List Completion.** Table 2 presents a comparison of the state-of-the-art sequential recommenders with our proposed CAGE variants on the list completion task. Based on the results, we can make the following observations. **Firstly**, for both autoregressive and non-autoregressive models, our proposed CAGE module can significantly improve the performance of the baseline models. For example, CAGE$_{BERT4Rec}$ can achieve an average improvement of 38% and 31% in terms of NDCG@5 and HR@5 among all datasets, compared with BERT4Rec. **Secondly**, since the FANS models leverage item category information in their design, they outperform other autoregressive baselines. However, our CAGE-integrated variant CAGE$_{FANS}$ can still achieve better performance than FANS, which implies that the end-to-end training models utilizing differentiable vector quantization can effectively learn improved clustering features compared to the word2vec+kmeans [30] approach that relies on pre-extracted features. **Thirdly**, in the Spotify dataset, the performance of the CNN-based Caser model is better than Transformer-based BERT4Rec model, which is aligned with the observation in [30]. One possible reason is that the local knowledge of the Spotify dataset is more important than the global information.

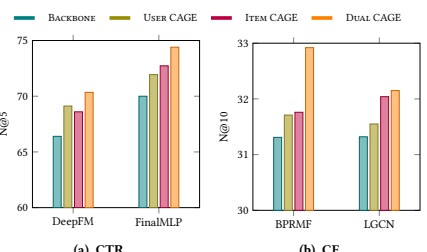

(a) CTR      (b) CF

**Figure 4: Influence of the use of user and item CAGE in the non-sequential recommenders.**

**Collaborative Filtering.** Table 3 displays the results of the popular CF models, along with our proposed CAGE variants on the collaborative filtering task. From the results, we can make the following observation. Our proposed CAGE consistently enhances the performance on the two datasets, resulting in significant improvements compared to the baseline models.

**Click-Through Rate Prediction.** Table 4 shows the results of the widely-used CTR prediction models and our proposed CAGE variants on the CTR prediction task. Based on the results, we can make the following observations. Among all CTR prediction models, our CAGE variants outperform the baseline models.

| Models | | DCN (2017) | CAGE_DCN (ours) | Imp. | DeepFM (2018) | CAGE_DeepFM (ours) | Imp. | FiBiNET (2019) | CAGE_FiBiNET (ours) | Imp. | FinalMLP (2023) | CAGE_FinalMLP (ours) | Imp. |
|---|---|---|---|---|---|---|---|---|---|---|---|---|---|
| MIND | N@5 | 0.2031 | **0.2173** | 6.99% | 0.2170 | **0.2428** | 11.89% | 0.2181 | **0.2319** | 6.33% | 0.2176 | **0.2265** | 4.09% |
| | N@10 | 0.2623 | **0.2770** | 5.60% | 0.2749 | **0.2992** | 8.83% | 0.2760 | **0.2881** | 4.38% | 0.2757 | **0.2823** | 2.39% |
| | HR@5 | 0.3958 | **0.4101** | 3.61% | 0.4065 | **0.4353** | 7.08% | 0.4081 | **0.4336** | 6.25% | 0.4061 | **0.4202** | 3.47% |
| | HR@10 | 0.5889 | **0.6012** | 2.09% | 0.5948 | **0.6156** | 3.49% | 0.5949 | **0.6129** | 3.03% | 0.5935 | **0.6007** | 1.21% |
| MovieLens | N@1 | 0.6781 | **0.7047** | 3.92% | 0.6640 | **0.7035** | 5.95% | 0.7016 | **0.7328** | 4.45% | 0.7000 | **0.7440** | 6.29% |
| | N@5 | 0.7029 | **0.7204** | 2.49% | 0.7014 | **0.7152** | 1.97% | 0.7314 | **0.7445** | 1.79% | 0.7337 | **0.7431** | 1.28% |
| | N@10 | 0.7465 | **0.7569** | 1.39% | 0.7433 | **0.7524** | 1.22% | 0.7679 | **0.7862** | 2.38% | 0.7696 | **0.7818** | 1.59% |
| | HR@5 | 0.9969 | **0.9984** | 0.15% | 0.9969 | **0.9984** | 0.15% | 0.9953 | **0.9969** | 0.16% | 0.9937 | **0.9987** | 0.50% |

**Table 4: Effectiveness of CAGE in click-through rate prediction. We bold the best results.**

| Datasets | | | Zhihu | | | | | | | | Goodreads | | | | | | | |
|---|---|---|---|---|---|---|---|---|---|---|---|---|---|---|---|---|---|---|
| Models | | | CAGE_BERT4Rec | | | | CAGE_FANS | | | | CAGE_BERT4Rec | | | | CAGE_FANS | | | |
| $v^1$ | $v^2$ | $v^3$ | N@5 | N@10 | HR@5 | HR@10 | N@5 | N@10 | HR@5 | HR@10 | N@5 | N@10 | HR@5 | HR@10 | N@5 | N@10 | HR@5 | HR@10 |
| 10 | - | - | 0.0197 | 0.0295 | 0.2151 | 0.3752 | 0.0280 | 0.0412 | 0.3002 | 0.4839 | 0.0113 | 0.0165 | 0.1654 | 0.2510 | **0.0396** | **0.0561** | **0.4267** | **0.5473** |
| 20 | - | - | **0.0208** | **0.0307** | **0.2327** | **0.3953** | 0.0273 | 0.0403 | 0.2895 | 0.4745 | 0.0105 | 0.0151 | 0.1654 | 0.2588 | 0.0388 | 0.0554 | 0.4120 | 0.5363 |
| 50 | - | - | 0.0199 | 0.0277 | 0.2204 | 0.3852 | 0.0299 | 0.0411 | 0.2975 | 0.4758 | **0.0119** | **0.0169** | 0.1673 | 0.2549 | 0.0375 | 0.0540 | 0.4066 | 0.5288 |
| 100 | - | - | 0.0142 | 0.0214 | 0.1691 | 0.3081 | 0.0288 | **0.0414** | 0.3044 | **0.4866** | 0.0104 | 0.0153 | 0.1511 | 0.2412 | 0.0366 | 0.0510 | 0.4034 | 0.5201 |
| 500 | - | - | 0.0057 | 0.0094 | 0.0910 | 0.1752 | 0.0249 | 0.0397 | 0.2702 | 0.4685 | 0.0110 | 0.0158 | 0.1654 | **0.2601** | 0.0367 | 0.0516 | 0.3865 | 0.5071 |
| 100 | 10 | - | 0.0192 | 0.0271 | 0.2129 | 0.3705 | 0.0270 | 0.0384 | 0.2884 | 0.4651 | 0.0096 | 0.0148 | 0.1654 | 0.2588 | 0.0387 | 0.0555 | 0.4021 | 0.5383 |
| 200 | 10 | - | **0.0235** | **0.0311** | **0.2429** | 0.3832 | 0.0271 | 0.0389 | 0.2970 | 0.4705 | **0.0130** | **0.0180** | 0.1829 | 0.2678 | 0.0368 | 0.0531 | 0.3872 | 0.5350 |
| 500 | 10 | - | 0.0223 | 0.0307 | 0.2376 | 0.3966 | **0.0301** | **0.0428** | 0.3034 | 0.4859 | 0.0108 | 0.0163 | 0.1673 | 0.2724 | 0.0379 | 0.0540 | 0.4092 | 0.5363 |
| 1000 | 10 | - | 0.0219 | 0.0308 | 0.2322 | 0.4040 | 0.0251 | 0.0364 | 0.2627 | 0.4436 | 0.0084 | 0.0119 | 0.1386 | 0.2121 | 0.0344 | 0.0491 | 0.3761 | 0.5032 |
| 200 | 20 | - | 0.0165 | 0.0244 | 0.1851 | 0.3275 | 0.0282 | 0.0412 | **0.3066** | 0.4899 | 0.0116 | 0.0159 | 0.1699 | 0.2536 | 0.0379 | 0.0542 | 0.3988 | 0.5318 |
| 400 | 20 | - | 0.0182 | 0.0276 | 0.2113 | 0.3799 | 0.0279 | 0.0412 | 0.2884 | **0.4919** | 0.0088 | 0.0133 | 0.1388 | 0.2348 | 0.0368 | 0.0538 | 0.3956 | 0.5337 |
| 8000 | 20 | - | 0.0197 | 0.0291 | 0.2349 | **0.4047** | 0.0268 | 0.0395 | 0.2900 | 0.4826 | 0.0089 | 0.0125 | 0.1420 | 0.2185 | 0.0373 | 0.0539 | 0.4092 | 0.5383 |
| 500 | 50 | - | 0.0114 | 0.0164 | 0.1482 | 0.2631 | 0.0294 | 0.0419 | 0.3028 | 0.4893 | 0.0112 | 0.0166 | 0.1783 | 0.2769 | **0.0399** | **0.0567** | 0.4275 | **0.5473** |
| 2500 | 50 | - | 0.0095 | 0.0148 | 0.1348 | 0.2517 | 0.0264 | 0.0388 | 0.2868 | 0.4631 | 0.0103 | 0.0148 | 0.1556 | 0.2510 | 0.0359 | 0.0516 | 0.3761 | 0.5065 |
| 4000 | 200 | 10 | 0.0199 | 0.0294 | 0.2301 | 0.3906 | 0.0266 | 0.0388 | 0.2895 | 0.4765 | 0.0090 | 0.0134 | 0.1446 | 0.2425 | 0.0393 | 0.0553 | 0.4040 | 0.5435 |
| 8000 | 400 | 20 | **0.0200** | 0.0285 | 0.2204 | 0.3691 | 0.0245 | 0.0365 | 0.2755 | 0.4651 | 0.0088 | 0.0136 | 0.1381 | 0.2425 | 0.0383 | 0.0544 | 0.4008 | 0.5350 |

**Table 5: Impact of the number of CAGE layers (H) and the number of entries of each layer ($v^i$). The best results are indicated in bold, while the second-best results are underlined. A hyphen (-) indicates the absence of a layer. For example, "100($v^1$) 10($v^2$) -($V^3$)" means that CAGE only has two layers, and the first and second layers correspond to the 100-entry and 10-entry codebooks, respectively. We fix $\alpha, \beta, \omega_c, \omega_q$ to be $1.0$ in this experiment.**

## 5.3 Ablation Study

**Structure of the Category Tree.** We study the effects of the number of layers and the number of entries (i.e., codebook size) in CAGE. We vary the number of layers from 1 to 3 and the number of entries within a range from 10 to 8,000. We fix other hyper-parameters and report the results of CAGE_BERT4Rec and CAGE_FANS. As illustrated in Table 5, we conduct experiments on the Zhihu and Goodreads datasets. From the results, we can make the following observations. **Firstly**, the best results of two-layer CAGE variants are better than those of one-layer CAGE variants on both datasets, indicating that CAGE can effectively capture the hierarchical category information to further improve the entity representations. **Secondly**, different variants prefer different numbers of entries. For example, on the Zhihu dataset, CAGE_BERT4Rec prefers a small number of entries in the first layer (i.e., 200), while CAGE_FANS prefers a large number of entries in the same layer (i.e., 500). **Thirdly**, different datasets prefer different numbers of entries. For example, for the CAGE_BERT4Rec variant, the best number of entries is 20 on the Zhihu dataset and 50 on the Goodreads dataset. **Fourthly**, as the number of entries increases, the performance of CAGE variants

first increases and then decreases. One possible reason is that a small number of entries may exhibit boundary effects, and as the entry size increases, the boundaries of the clusters gradually become blurred. However, when the number of entries is too large, the number of entities in each entry is too small, which may lead to insufficient learning of categorical feature. Moreover, the layer and entry numbers need to be carefully adjusted, otherwise it may lead to negative effects.

**Effectiveness of Item/User CAGE.** We also test the effectiveness of the dual CAGE (i.e., using both user and item CAGE) in both CF and CTR prediction scenario. As shown in Figure 4, the results prove that both user and item CAGE could boost the performance of baselines.

## 5.4 Impact of Hyper-parameters

We explore the impacts of the residual connection weight $\alpha$, the quantization commitment cost $\beta$, the quantization loss weight $\omega_q$, and the codebook classification loss weight $\omega_c$. The experiments are conducted on two list completion datasets, i.e., Zhihu and Goodreads. Based on the results from Section 5.3, we take the best CAGE configuration of the CAGE_FANS model, i.e., (500, 10) for

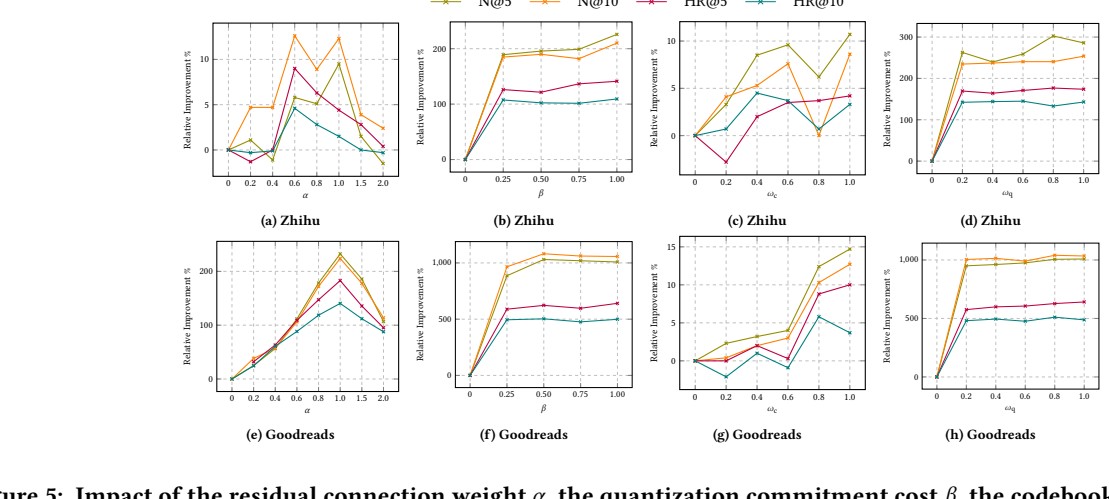

**Figure 5: Impact of the residual connection weight $\alpha$, the quantization commitment cost $\beta$, the codebook classification loss weight $\omega_c$, and the quantization loss weight $\omega_q$. We use the model with $\alpha = 0$ as the reference baseline for (a), and measure the *relative improvement* of each metric compared to the baseline for various values of $\alpha$, defined as $(m_\alpha - m_0)/m_0 * 100\%$, where $m$ is one of the metrics in {N@5, N@10, HR@5, HR@10}. Therefore, the relative improvement of $\alpha = 0$ is constant at 0%. Similarly, we use the model with $\beta = 0$ as the reference baseline for (b), $\omega_q = 0$ for (c), and $\omega_q = 0$ for (d).**

the Zhihu dataset and (500, 50) for the Goodreads dataset. Based on the results from Figure 5, we can make the following observations. **Firstly**, the performance of baselines (i.e., when hyper-parameters are set to 0) is inferior to the most of the cases, indicating the effectiveness of these hyper-parameters. **Secondly**, different datasets achieve the best performance at different hyper-parameter settings. For example, the Zhihu dataset reaches the best performance at $\alpha = 0.6$, while for the Goodreads dataset, $\alpha = 1.0$. **Thirdly**, unlike the computer vision domain where the quantization commitment cost $\beta$ is usually set to 0.25 [46], in the recommendation domain, a higher $\beta$ (i.e., 1.0 for the Zhihu dataset or 0.50 for the Goodreads dataset) gets a higher performance. **Fourthly**, due to the equivalent performance shown by the hyperparameters when set to 1 in the Figure 5(a)(b)(c)(d) (e.g., the performance of $\alpha = 1$ in Figure 5(a) is equivalent to that of $\beta = 1$ in Figure 5(b)), we can assess the performance when the hyperparameters are set to 0 by examining the range on the vertical axis (e.g., comparing the performance of $\alpha = 0$ in Figure 5(a) with that of $\beta = 0$ in Figure 5(b)). A wider range signifies a larger disparity between the performance at 0 and 1 for the hyperparameters. This indicates that when this particular hyper-parameter is set to 0, the resulting effect is poorer, highlighting its greater significance. Therefore, we can observe that the ranking of importance for these four hyperparameters is: $\omega_q > \beta > \alpha \approx \omega_c$. Similarly, according to Figure 5(e)(f)(g)(h) for the Goodreads dataset, the importance ranking is: $\omega_q \approx \beta > \alpha > \omega_c$.

## 5.5 Visualization

Here, we demonstrate the clustering quality of our CAGE. We take the MIND dataset which has ground truth category labels in the experiments. We set the levels of vector quantizer to 1 (i.e., $H = 1$) and the codebook size to 20 (i.e., $v^1 = 20$).

Before training (we use the DeepFM model as the backbone), we randomly select one real category. After each training epoch, we

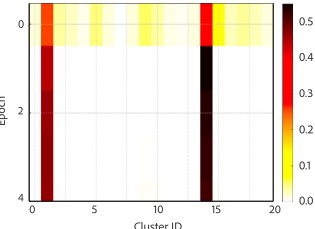

**Figure 6: Visualization of the learned categorization.**

calculate the relative proportion of news articles that each codebook entry contains for the current category. This operation will yield an array of 20 numbers, and their sum equals one. After training, we aggregate the arrays collected at the end of each epoch into a two-dimensional array and create a HeatMap as shown in Figure 6. We can observe that news articles for the current category are dispersed among different clusters. As training progresses, these news articles quickly converge into two clusters and stabilize in the subsequent phases, which demonstrates the effectiveness of the categorization of our CAGE.

## 6 CONCLUSION

We have proposed CAGE, a novel framework for leaning item/user category trees for ID-based recommendation, by employing differentiable vector quantization techniques. The flexibility of CAGE allows for its seamless integration into a variety of existing recommender systems. Through comprehensive experiments conducted across diverse recommendation scenarios, we have demonstrated the effectiveness of CAGE in enhancing the performance of various recommendation models. Additionally, our visualization experiments have further validated the robustness of the learned categorical knowledge.

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

Received 20 February 2007; revised 12 March 2009; accepted 5 June 2009

