# OpenReview forum: "Learning Category Trees for ID-Based Recommendation: Exploring the Power of Differentiable Vector Quantization"
_ACM.org/TheWebConf/2024/Conference — TheWebConf24_

### Official Review · Reviewer_V8jg · 2023-11-23

**Novelty:** 4
**Technical Quality:** 4

**Review:**

Strengths: This paper proposes the CAGE, an end-to-end category tree generation framework for ID-based recommendation. Extensive experimental results prove the effectiveness of the method. Overall, the paper is easy to follow.

Weaknesses: 1.This paper, especially the mathematical expression, is somewhat difficult to understand. For example, in Section 3.1, "Given an input embedding e , nearest neighbour search is performed to find the most similar code to z within C", what is z, is this a typo mistake? Moreover, in Section 3.2, q is not defined after it is first used. There are two same symbols in Equation 11.
2.Although the results demonstrate the effectiveness of CAGE across most scenarios, will the Integration of the proposed module have a significant impact on the efficiency of the model, especially on large-scale dataset? It would be better to have deeper analysis.

**Questions:**

Questions are similar to the weaknesses listed above.

**Reviewer Confidence:**

3: The reviewer is confident but not certain that the evaluation is correct

**Scope:**

3: The work is somewhat relevant to the Web and to the track, and is of narrow interest to a sub-community

---

### Official Review · Reviewer_39Bg · 2023-11-23

**Novelty:** 5
**Technical Quality:** 5

**Review:**

## summary:

The paper introduces a pioneering automated category tree generation
(CAGE) via differentiable vector quantization framework. This
framework enables the simultaneous learning and refinement of
categorical code representations and entity embeddings in an
end-to-end manner.

## strengths:

- s1. The narrative and explanations are skillfully crafted, ensuring
  clarity and ease of comprehension.

- s2. The framework exhibits seamless adaptability to various recommendation
  system tasks, including List Completion, Click-Through Per Rate, and
  Collaborative Filtering, as well as compatibility with diverse methods
  such as GRU4Rec, Bert4Rec, NeuMF, and others.

- s3. Certain methods within the framework demonstrate a noteworthy
  enhancement in performance across diverse tasks.

## Weaknesses:

- w1. The purpose of having Vector Quantization granularity is unclear when
  the vectors are ultimately averaged at the end of the process.

- w2. The explicit connection between Tree Generation and Vector
  Quantization is not clearly delineated, leaving room for ambiguity in
  understanding the relationship between these components.

- w3. The paper lacks a clear description of the FANS models employed,
  making it challenging for readers to grasp the specifics of the models
  used in the study.

- w4. There is a significant disparity in the performance of the proposed
  method, notably excelling for CFKG and BERT4Rec, while showing minimal
  impact on others like LGCN and NeuMF. An elucidation on why such
  variations exist would enhance the paper's comprehensibility.


suggestion for improvement:

- In Figure 5, it is recommended to either increase the marker size or
  modify the colors of the markers, as the current use of yellow and
  green makes them difficult to distinguish.

- Enhance the statistical robustness and reliability of the
  experiments by providing insights into standard deviation and
  p-values. This additional information will contribute to a more
  comprehensive understanding of the results.

- Consider conducting an additional ablation study that significantly
  increases the number of quantilizers (H). Thus far, it has been
  observed on the Table 5 that higher numbers yield better
  results. Investigate whether there is a tipping point at which the
  results plateau or sharply deteriorate.

- Add an "Improvement" column to Table 2 specifically for non-FANS
  models. The origin of the reported percentages (31% and 38%) in
  Section 5.2 List Completion is unclear, and including an improvement
  column will provide transparency and clarity regarding these values.

**Questions:**

see weaknesses above

**Reviewer Confidence:**

3: The reviewer is confident but not certain that the evaluation is correct

**Scope:**

3: The work is somewhat relevant to the Web and to the track, and is of narrow interest to a sub-community

---

### Official Review · Reviewer_YYB8 · 2023-11-23

**Novelty:** 7
**Technical Quality:** 6

**Review:**

#Summary

This paper presents a novel approach for automatically generating category trees using vector quantization. The method utilizes only user/item ID to derive a hidden categorical structure, which is then used as additional information to enhance the quality of recommendations. This is an end-to-end process that does not rely on any external information. Furthermore, this approach is adaptable to various recommendation tasks, such as classic CF, CTR, and List Completion. The experiments conducted in this study are extensive, including seven datasets and 14 baseline models.

#Clarity, Originality, and Significance

The paper is well-written and technically sound. However, I am not familiar with using vector quantization in recommender systems. From what I can tell, this approach appears to be original, which makes it significant for the field, with a high potential for academic impact.

#Pros

- The paper presents an elegant solution to a relevant and open research problem;
- The method is clearly described, and the authors promise to share their code upon paper publication;
- The experiments are extensive, including a large set of datasets, baselines, and results analyses.

#Cons

- There are several categories of Amazon datasets available. Why these particular ones? There are more commonly used benchmarks for CF than the ones chosen by the authors, e.g., the MovieLens suite. For example, see "Steffen Rendle, Walid Krichene, Li Zhang, and Yehuda Koren. 2022. Revisiting the Performance of iALS on Item Recommendation Benchmarks. In Proceedings of the 16th ACM Conference on Recommender Systems (RecSys '22)". A similar comment goes to the datasets of the CTR task. Despite being very small (MovieLens 100K is considered a toy dataset nowadays), there are datasets more suited for this particular task;
- It is unclear how the baselines were fine-tuned. The authors mention that certain values were used for all members of a specific method family, such as the use of three hidden layers for Transformer-based models. However, they also claim that the hyperparameters of all models were carefully fine-tuned. It would be helpful if the authors could specify which hyperparameters were fine-tuned and which ones were not;
- For each task, comparing against methods that leverage explicit user/item categories would be a nice addition since most of the baselines do not use any kind of additional data;
- I missed a deeper analysis of how the learned categories resemble real ones. Section 5.5 gives some clues but still very preliminary.

**Questions:**

How do the learned categories resemble real ones?

**Reviewer Confidence:**

3: The reviewer is confident but not certain that the evaluation is correct

**Scope:**

4: The work is relevant to the Web and to the track, and is of broad interest to the community

---

### Official Review · Reviewer_9cxE · 2023-11-24

**Novelty:** 4
**Technical Quality:** 3

**Review:**

This manuscript proposes a differentiable vector quantization framework for automatic category tree generation (CAGE), aiming to implicitly drive the encoding of category information from IDs to enhance ID-based recommendations. The effectiveness of CAGE is verified on three recommendation scenarios and 7 public data sets.

Pros:
* How to use vector quantization to improve recommendations more effectively is a valuable research direction.
* The solutions proposed are easy to understand.
* Provides rich experimental results.

Cons:
* The need to introduce vector quantization is not discussed enough.
* There are some limitations on the plausibility verification of solutions.
* There are some limitations to the compatibility verification of solutions.

Below is a further description of the Cons:
* The current description of the motivations for adopting VQ as a core component in the solution is somewhat inadequate. On the one hand, the authors transition directly to the description of CAGE by discussing the limitations of traditional three-stage vector quantization pipelines when only IDs are available. On the other hand, the core idea of the manuscript lies in supervising a clustering process of ID information through external recommendation loss. One question is whether other strategies implementing this clustering process would be better than VQ, such as traditional data-driven clustering methods.

* Since implicitly driving category information from IDs is not so intuitive, it is necessary to design some experiments to show that CAGE can indeed capture the real category information. For example, use a dataset with some class information and mask them during training. In addition, it is also necessary to explore the semantics of beneficial category information captured by CAGE in different architectures by combining the obtained category encoding with the real category information in the dataset.

* To further emphasize the solution's effectiveness, I think a compatibility experiment is also necessary, i.e., applying the category encoding of CAGE trained based on the A architecture to other architectures for performance evaluation.

**Questions:**

Please see the comments in the Review section.

**Ethics Review Description:**

N/A.

**Reviewer Confidence:**

4: The reviewer is certain that the evaluation is correct and very familiar with the relevant literature

**Scope:**

3: The work is somewhat relevant to the Web and to the track, and is of narrow interest to a sub-community

---

### Official Review · Reviewer_pygN · 2023-11-29

**Novelty:** 5
**Technical Quality:** 6

**Review:**

This paper proposes a novel approach called CAGE for automatically learning and generating entity category trees for ID-based recommendation. The authors use a differentiable vector quantization (VQ) technic for category tree generation, which enables the simultaneous learning and refinement of categorical code representations and entity embeddings in an end-to-end workflow. The CAGE approach is highly adaptable and can be easily integrated into both sequential and non-sequential recommender systems.


  ### **PROS**

- **Novelty and empirical validation**
  - The novelty relies in the proposed CAGE approach, which , it also lies in its empirical validation of the effectiveness of the CAGE approach on various recommendation tasks, including list completion, collaborative filtering, and CTR prediction, across different recommendation models
- **Comprehensive evaluation**
  - The authors provide some information about the datasets including several real-world datasets, preprocessing steps, and evaluation metrics used in their experiments. And they also mentioned that more details about the datasets and their reprocessing will be available in the public code repository upon acceptance. it provides strong evidence for the usefulness of the proposed method CAGE.
- **Detailed ablation studies**
  - The paper includes detailed ablation studies that analyze the effects of number of layers and the number of entries in CAGE while fixing other hyper-parameters. The authors also provided a table that summarize the results in table clearly. The ablation studies suggest that increasing the number of layers and entries in CAGE can improve the performance of recommendation models within certain threshold, however, beyond the threshold leads to diminishing returns.

 ### **CONS**

- **Lack of implementation details and instruction for reproduction**
  - The paper doesn't provide detailed implementation information and instructions for reproducing the experiments though there will be public code will be available upon the acceptance. It may makes it difficult for other researchers to verify the results and build the proposed approaches since the experiments are complicated.
- **Other limitations**
  - The paper assumes the category tree is known or can be automatically generated, which may not always be the case in practice and other certain scenarios.
  - This paper focuses on ID-based recommendation, which may not be applicability to other types of recommendation tasks or domains and limits the generalizability of the CAGE.

**Questions:**

1. How does the CAGE approach compare to other state-of-the-art deep CTR models in terms of performance on benchmark datasets? And more details?
2. What are the advantages of using differentiable VQ for learning categorical knowledge in ID-based recommendation, and how does it compare to other techniques used in automatic category tree generation?
3. Does the CAGE can be extended to handle multi-modal or heterogeneous data source, such as text, image or video? If yes, can it capture the semantic relationships between different modalities?
4. How does the CAGE approach address these limitation mentioned above?

**Reviewer Confidence:**

3: The reviewer is confident but not certain that the evaluation is correct

**Scope:**

3: The work is somewhat relevant to the Web and to the track, and is of narrow interest to a sub-community

---

### Decision · Program_Chairs · 2024-01-22

**Decision:**

Accept

**Comment:**

The paper proposes a method for automatically generating category trees for improving recommender system prediction quality.

 The reviewers see the paper as overall well written with minor suggestions for improvements. The reviewers like that the method can be applied to a broad class of models and recommendation tasks. Another strength of the paper is that it evaluates the quality improvements on a variety of tasks.

 The reviewers also pointed out limitations. Among them:

 * A limitation of the proposed work is that it does not investigate sufficiently why VQ is the right method for learning trees nor how good the quality of the learned trees is. The authors provided some more results in this direction at the very end of the discussion period.
 * There were some concerns with the experimental setup and the lack of reusing established benchmarks where published results exist. The authors provided experiments on more datasets at the end of the rebuttal, however the authors did not reuse established splits, so it is not possible to compare their results to published ones.

 Some rebuttals were provided on the final day which is too late for the reviewers to engage in a discussion with the authors.

 Overall, this is interesting work. There is some room for improvement, especially with respect to investigating the quality of the generated trees and substantiating the choice of VQ for tree learning. I see the submitted work as a borderline paper slightly leaning towards acceptance.